# Co-Targeting MAP Kinase and Pi3K-Akt-mTOR Pathways in Meningioma: Preclinical Study of Alpelisib and Trametinib

**DOI:** 10.3390/cancers14184448

**Published:** 2022-09-13

**Authors:** Gregoire Mondielli, Gregory Mougel, Florent Darriet, Catherine Roche, Adeline Querdray, Christophe Lisbonis, Romain Appay, Henry Dufour, Olivier Chinot, Thomas Graillon, Anne Barlier

**Affiliations:** 1Aix Marseille Univ, INSERM, MMG, UMR1251, Marmara Institute, 13855 Marseille, France; 2Aix Marseille Univ, INSERM, MMG, UMR1251, Marmara Institute, APHM, La Conception Hospital, Laboratory of Molecular Biology, 13855 Marseille, France; 3APHM, La Conception Hospital, Laboratory of Molecular Biology, 13385 Marseille, France; 4APHM, CHU Timone, Service d’Anatomie Pathologique et de Neuropathologie, Aix-Marseille Univ, CNRS, INP Inst Neurophysiopathol, 13385 Marseille, France; 5Marseille Univ, INSERM, MMG, UMR1251, Marmara Institute, APHM, La Timone Hospital, Neurosurgery Department, 13385 Marseille, France; 6APHM, La Timone Hospital, Neurooncology Department CNRS, INP Inst Neurophysiopathol, 13855 Marseille, France; 7Department of Neurosurgery, La Timone Hospital, APHM, 264 Rue Saint-Pierre, 13005 Marseille, France; 8Marseille Medical Genetics, Faculté Timone, 27 Bd Jean Moulin, CEDEX 05, 13855 Marseille, France

**Keywords:** meningioma, *NF2*, targeted therapy, alpelisib, trametinib, everolimus, MAPkinase, Pi3kinase

## Abstract

**Simple Summary:**

Multi recurrent or high-grade meningiomas remain an unmet medical need in neuro-oncology. Several studies have highlighted the potential therapeutic efficacy of mTor inhibitors to control tumoral growth of meningiomas. However, a positive feedback on AKT oncogenic pathway from these drugs may explain the modest success. Our aim was to target Pi3kinase upstream mTor, and MAP kinase pathway, overactivated in meningiomas, alone or in combined targeting in comparison to mTor targeting. Our in vitro results obtained on three meningioma cell lines and on a large series of fresh human meningiomas, including 35 WHO grade 1, 23 grade 2, and five grade 3, showed that co-targeting Pi3kinase and MAP kinase seemed promising, opening new therapeutic strategies in these tumors.

**Abstract:**

Recurrent or high-grade meningiomas are an unmet medical need. Recently, we demonstrated that targeting mTOR by everolimus was relevant both in vitro and in humans. However, everolimus induces an AKT activation that may impact the anti-proliferative effect of the drug. Moreover, the MAP kinase pathway was shown to be involved in meningioma tumorigenesis. We therefore targeted both the Pi3k-AKT-mTOR and MAP kinase pathways by using combinations of the Pi3k inhibitor alpelisib and the MEK inhibitor trametinib. Our study was performed in vitro on the human meningioma cell lines and on a large series of primary cultures providing from 63 freshly operated meningiomas including 35 WHO grade 1, 23 grade 2, and five grade 3, half of which presented a *NF2* genomic alteration. Alpelisib induced a higher inhibitory effect on cell viability and proliferation than everolimus in all cell lines and 32 randomly selected tumors no matter the genomic status, the histological subtype or grade. Trametinib also strongly inhibited cell proliferation and induced AKT activation. Combined treatment with alpelisib plus trametinib reversed the AKT activation induced by trametinib and induced an additive inhibitory effect irrespective of the cell lines or tumor features. Co-targeting pathways seems promising and may be considered particularly for aggressive meningioma.

## 1. Introduction

Recurrent or high-grade meningiomas are an unmet medical need in neuro-oncology [1]. The six-month progression-free survival rate (PFS6) of aggressive refractory WHO (World Health Organization) grade 2–3 meningioma is only 10–15% without treatment [2]. Anti-VEGF drugs sunitinib and bevacizumab have brought rare cases to PFS6 of respectively 42% and 41%. However, the effects were transient [3,4].

Although mutations in components of the phosphoinositide-3-kinase/AKT/mammalian target of rapamycin (Pi3K-AKT-mTOR) signaling pathway are rare and mostly observed in non-aggressive meningioma [5], we have demonstrated that targeting this pathway may be relevant in aggressive meningioma, supporting the clinical use of the mTOR inhibitor, everolimus [6,7,8,9]. In fact, Pi3k-AKT-mTor is overactivated in the majority of meningioma [10]. Earlier preclinical results highlighted the anti-proliferative effects of everolimus and the somatostatin analog, octreotide [11]. Combinations of both drugs provided an additive anti-proliferative effect in vitro [11,12]. Clinical trial CEVOREM (NCT02333565), testing a combination of both drugs, yielded a PFS6 at 55% with a decrease in tumor growth rates and volume stabilization in numerous cases [13,14]. However, we have found that AKT is activated in response to everolimus in vitro, which can decrease the anti-proliferative effect of the drug [11]. We hypothesized that targeting Pi3K upstream of mTOR could prevent this positive feedback. Alpelisib (BYL719) is a Pi3K p110α-specific inhibitor already used for breast cancer [15].

We have never observed apoptosis under everolimus or octreotide in our in vitro studies on meningiomas suggesting that only an antiproliferative effect was induced by these drugs in vivo [11,12]. We hypothesized that an additional apoptotic effect may reduce tumor mass and therefore the drug response. The mitogen activated protein (MAP) kinase pathway activation was demonstrated in all meningioma, no matter the WHO grade and despite a lack of mutations identified in components of the pathway [8,16,17]. Moreover, in vitro, targeted inhibition of this pathway induces not only an anti-proliferative effect but also an apoptotic effect in meningioma cells [8]. Trametinib, a MEK inhibitor, is currently used in combined treatment for recurrent melanomas or lung cancer in clinical practice at 1–2 mg daily [18,19]. These data strongly suggest that it would be promising in aggressive or recurrent meningioma to target either alone or in combination, the Pi3K-AKT-mTOR pathway with the Pi3K inhibitor alpelisib and the MAP kinase pathway with the MEK inhibitor trametinib. We therefore tested these drugs, alone and in combination, firstly on three human meningioma cell lines, CH-157MN, IOMM-lee, and Ben-Men-1, and on a large series of primary cell cultures originating from 63 freshly operated meningiomas, well characterized both clinically and at the genomic level. We analyzed cell viability and proliferation, apoptosis, and the main components of the MAP kinase and Pi3K-AKT-mTOR transduction pathways.

## 2. Materials and Methods

### 2.1. Materials

Alpelisib, everolimus, and trametinib were obtained from Novartis International AG (Basel, Switzerland). Drugs were dissolved in 100% dimethylsulfoxide (DMSO) to a 10 mM concentration and stored at −80 °C before dilution to intermediate concentrations in cell culture medium. In all the pharmacological experiments, control cells were treated with an equivalent 0.1% vehicle DMSO concentration, equivalent to the 0.1% final DMSO concentration in the drug dilution.

### 2.2. Cell Culture of 3 Meningioma Cell Lines

This study was carried out on three human meningioma cell lines, Ben-Men1, CH-157MN and IOMM-Lee. The Ben-Men-1 comes from WHO grade 1 meningothelial meningioma (accession number CVCL_1959, Braunschweig, Germany), CH-157MN comes from meningioma with unknown WHO grade (accession number CVCL_5723kindly given by CERIMED), IOMM-Lee comes from WHO grade 3 bone tumor invasion of a meningioma (accession number CVCL_5779, ATCC, Manassas, VA, USA). The three cell lines were cultured in DMEM-F12 (Dulbecco’s Modified Eagle Medium) 10% FCS (fetal calf serum), 1% streptomycin/penicillin, 1% fungizone (Invitrogen, Cergy Pontoise, France), at 37 °C with 7% CO_2_.

### 2.3. Primary Cell Culture of Freshly Excised Human Meningioma

This study was carried out in human meningiomas from 63 patients (Appendix A). Tumor grading was done according to the 2021 WHO classification of CNS tumors criteria. The present study was undertaken after informed consent was obtained from each patient. Briefly, freshly excised tumor fragments were minced into pieces smaller than 1 mm^3^ and disaggregated into single cells by exposure to 0.37% type I collagenase (Invitrogen, Cergy Pontoise, France) for 2 h at 37 °C in culture medium (the same than that of cell lines). The cells were cultured in complete medium (10% of FCS) and the experiments were performed in the first 3 weeks after surgery to preserve differentiated features of tumoral cells as previously described [11,12]. Experiments were performed according to the quantity of tumor cells available after tumor dissociation. In each functional in vitro experiment, tumors were randomly selected according to the available cell number obtained after tumor dissociation as they arrived (see Appendix A).

### 2.4. Genomic Characterization

DNA was extracted from frozen fragments of meningiomas and from cell pellets for human cell lines using the QIAamp DNA kit (Qiagen, Courtaboeuf, France). DNA was analyzed by sequencing on a MiSeqDx (Illumina, San Diego, CA, USA), using the Custom QIAseq Targeted DNA Panel library preparation (Qiagen, Courtaboeuf, France) targeting 13 meningioma genes: *NF2* (NM_000268.3), *AKT1* (NM_001014431.1), *SMO* (NM_005631.4), *KLF4* (NM_004235.4), *TRAF7* (NM_032271.2), *PIK3CA* (NM_006218.4), *SUFU* (NM_016169.4), *SMARCB1* (NM_003073.3), *SMARCE1* (NM_003079.4), *CDKN2A* (NM_058195.3), *CDKN2B* (NM_004936.3), *PTEN* (NM_000314.4) and *TERT* promoter (NM_198253.2) [20]. The alignment and the variant calling were performed using 2 pipelines: the CLC Genomics Workbench 20.0.4 (Qiagen) and the QIAseq targeted DNA custom panel analysis center (smCounter2 Qiagen). Each variant was classified in the five classes consistent with the guidelines of the American College of Medical Genetics and Genomics (ACMG) [21]. Literature data, COSMIC (Catalogue of Somatic mutations in Cancer, https://cancer.sanger.ac.uk/cosmic, accessed on 3 June 2021) and VarSome (https://varsome.com/, accessed on 3 June 2021), were used to classify the variants. Moreover, canonical *BRAF* (NM_004333), *KRAS* (NM_002524), or *NRAS* (NM_004985), mutation was checked in the 3 cell lines and in all tumors by Sanger sequencing.

To search for large deletions or duplications of the *NF2* gene, quantitative real-time PCR was used, using a TaqMan™ Copy Number Assay (ref Hs00918833_cn, Applied Biosystem, Foster City, CA, USA) and a TaqMan™ Copy Number Reference Assay (RNase P) on an Applied Biosystems™ real-time PCR instrument (ViiA 7 Real-Time PCR System, Thermo Fisher Scientific Inc., Waltham, MA, USA). The data were analyzed using the Applied Biosystems Copy Caller software (Applied Biosystem, Foster City, CA, USA). Each sample was standardized using a *NF2* biallelic somatic DNA as calibrator and the relative quantification (RQ) of *NF2* was measured under the formula 2^−δδCt^ as compared *PRORP* (protein only RNase P catalytic subunit) as a standard gene. A RQ between 0.3 and 0.7 was interpreted as a deletion and a RQ higher than 0.8 was considered as normal (Appendix A).

### 2.5. Cell Viability

Cell viability was assayed by a luminescent cell viability assay (Cell Titer Glo, Promega Corporation, Charbonnier, France) in triplicate wells containing 2 × 10^4^ meningioma cells (see Appendix A). Twenty-four hours after plating, the cells were incubated in low serum media (5% FCS) and treated with drugs alone or in combination for 3 days. The results are expressed as the mean percentage of cell viability in treated vs. non-treated cells.

### 2.6. BrdU (5-Bromo-2′-Deoxyuridine) Incorporation

Four thousand cells were plated in a 96-well plate. After 24 h, the cells were incubated in low serum media and treated with drugs for 2 days. On the third day, BrdU was added to a final concentration of 1 µM. After incubation for 16 h, DNA synthesis was assayed with the Cell Proliferation ELISA BrdU kit (Roche Molecular, Biochemical, Meylan, France). Newly synthesized BrdU-DNA was determined using an Enspire Multimode plate reader (Perkin Elmer, Villebon-sur-Yvette, France).

### 2.7. Apoptosis Analysis

Apoptosis was studied through PARP (Poly(ADP)-ribose polymerase) cleavage in Western blot analysis (see below) and assessment of caspase 3 and 7 activities with a luminescent Caspase-Glo 3/7 Assay (Promega Corporation, Charbonnier, France). Caspase activities were measured in quadruplicate 6-well plates, 24 h after seeding 5 × 10^4^ primary meningioma cells per well. Cells were incubated in low-serum media (5%) and then treated with drugs for 2 days. Three hours of staurosporine (Sigma-Aldrich, St Quentin Fallavier, France) treatment (10^−6^ M) was used as positive control.

### 2.8. Western Blot Analysis

Intra-tumoral ERK and phospho-ERK levels were determined in frozen tumor fragments. The impact of drugs on signal transduction pathways and on apoptosis (PARP cleavage) was analyzed on cell cultures of 5 × 10^5^ primary meningioma cells, seeded into 6-well plates. After 24 h cells were incubated in low serum media (5%) and treated with drugs for different time according to the experiments. Cells were washed with ice-cold PBS and scraped in 100 µL of 25 mM HEPES-NaOH pH 7.4, 150 mM NaCl, 1% NP40 in the presence of protease inhibitors (2 mM sodium orthovanadate, 1 mM sodium fluoride, 10 mM beta-glycerophosphate, 10 µg/mL aprotinin and 10 µg/mL leupeptin). All reagents were purchased at Euromedex (Souffelweyersheim, France). At 4 °C, lysates were first agitated for 15 min (min) then centrifuged 20 min at 10,000× *g*. Protein concentration was determined by the DC Protein assay II (Bio-Rad, Marnes-La-Coquette, France) using an Enspire Multimode plate reader (PerkinElmer, Villebon-sur-Yvette, France) according to the manufacturer’s protocol. Proteins were diluted in NuPAGE LDS Sample buffer (Invitrogen, Cergy Pontoise, France) to 1–5 µg/µL.

After extraction, proteins were separated by 10% or 8% SDS-PAGE. Samples (20 µg/lane) were heated 5 min at 100 °C under reducing and denaturing conditions with 20 mM DTT and 2.5% (*w*/*v*) SDS. Proteins were transferred onto 0.45 µm PVDF Immobilon-P membrane (Millipore, Saint-Quentin-en-Yvelines, France), using the Pierce Power Station (ThermoFisher Scientific, Illkirch-Graffenstaden, France) according to the manufacturer’s protocol. Membranes were blocked with TBS and 0.005% Tween 20 (TBST) containing 5% BSA (1 h at room temperature) then incubated with primary antibodies (2 h at room temperature): rabbit polyclonal anti-phospho (Thr202/Tyr204) ERK1/2 (Cell Signaling Technology #9101), rabbit polyclonal anti-ERK1/2 (Cell Signaling Technology #9102), rabbit polyclonal anti-PARP (Cell Signaling Technology #9542), rabbit monoclonal anti-phospho (Ser235/236) S6 Ribosomal Protein (Cell Signaling Technology #4857), mouse monoclonal anti-S6 Ribosomal Protein (Cell Signaling Technology #2317), rabbit polyclonal anti-AKT (Cell signaling #9272), rabbit polyclonal anti-phospho (Ser473) AKT (Cell Signaling Technology #9271), or mouse monoclonal anti-GAPDH (Merck Millipore #MAB374, Molsheim, France). All Cell Signaling Technologies antibodies were purchased at Ozyme (Saint-Cyr-l’Ecole, France). Blots were developed with the Super Signal West Pico or femto PLUS Chemiluminescent Substrate (ThermoFisher Scientific, Illkirch-Graffenstaden, France). Chemiluminescent signals were detected using a CCD camera and Syngene software (G:BOX, Ozyme, Saint-Cyr-l’Ecole, France). Signal vs. background was quantified using NIH Image J #1.53f51 software (imagej.nih.gov/ij/, accessed on 3 June 2022).

### 2.9. Statistical Analysis

Results are presented as mean ± SEM (Standard Error of Mean). The statistical significance between two unpaired groups was determined by the Mann–Whitney non-parametric test and between two paired groups by the Wilcoxon non-parametric test. To measure the strength of association between pairs of variables without specifying dependency, Spearman rank order correlations were run. Differences were considered significant at *p* < 0.05. For contingency tables, a chi-squared test was performed. IC_50_ is half maximal inhibitory concentration. Regarding IC_50_ and maximal inhibition calculation, statistical analyses were performed using PRISM GraphPad 9 software (San Diego, CA, USA).

## 3. Results

### 3.1. Genomic Characterization of the Three Human Meningioma Cell Lines

The genomic analysis showed (i) in Ben-Men-1: a complete loss of one *NF2* copy associated to the *NF2* truncated mutation c.640delC, p.(Leu214fs*6), (ii) in CH-157MN: a complete deletion of one *NF2* copy associated with a *NF2* mutation in a splicing site c.241-1G>T, the well-known activating p.(Gln61Lys) (c.181C>A) of *NRAS,* the truncating variant p.Leu57Trpfs*42 (c.170delT) of *PTEN* and the mutation c.124C>T in the *TERT* promoter, (iii) in IOMM-Lee: the lack of alteration in *NF2* gene, the well-known activating mutation p.Val600Glu (c.1799T>A) of *BRAF* and the mutation c.124C>T in the *TERT* promoter. As expected, no *NF2* protein, Merlin, was seen by western-blotting in Ben-Men-1 and CH-157MN in contrast to IOMM-Lee (Appendix A).

### 3.2. Drugs Effects on Signaling Pathways and Cell Viability on the 3 Meningioma Cell Lines

To assess the target of each drug, the Pi3K-AKT and ERK pathways were checked in the three cell lines (Figure 1). Alpelisib inhibited the phosphorylation levels of AKT and had no impact on the ERK pathway. Trametinib alone or in combined treatment with alpelisib or everolimus inhibited the phosphorylation levels of ERK. As expected, everolimus alone or in combined treatment with alpelisib or trametinib inhibited the phosphorylation levels of pS6. The combined treatment trametinib plus everolimus increased the phosphorylation level of AKT.

Alpelisib inhibited cell viability of the three cell lines with the same manner (mean maximal effect 91 ± 6% at 10^−4^ M, IC_50_ 9.6 × 10^−6^ M, Figure 2A). The maximal inhibitory effect was stronger than those under everolimus (mean maximal effect 45 ± 10% at 10^−6^ M, IC_50_ 7.1 × 10^−10^ M, Figure 2B). Under everolimus, a slightly but significantly lower effect was observed in the wt *NF2* cell line, IOMM-Lee, than in the mutated *NF2* ones, Ben-Men-1 and CH157-MN (*p* < 0.01). Trametinib inhibited cell viability in the 3 cell lines. In agreement with the presence of *BRAF* mutation in IOMM-Lee, the IC_50_ was lower in this cell line in comparison with the 2 others CH-157-MN and Ben-Men-1 (0.1 × 10^−8^ M vs. 1 × 10^−8^ M and 2 × 10^−8^ M respectively). The mean maximal effect of the drug was in the three cell lines 60 ± 3%, 33 ± 2%, and 28 ± 6%, respectively (Figure 2C).

Then several combined treatments were explored at the concentration closed to the IC_50_ of each drug. A stronger effect was observed relative to each drug alone for alpelisib plus trametinib in Ben-Men-1 and IOMM-Lee (*p* < 0.0001) but not in CH-157MN. This effect was similar to the calculated sum of inhibitory effects for both drugs suggesting an additive effect (Figure 2D). A stronger effect of trametinib plus everolimus or of alpelisib plus everolimus relative to each drug alone was also observed (Appendix A).

### 3.3. Clinical and Genomic Characterization of the Human Meningioma

Sixty-three meningiomas were included, coming from 27 men and 36 women aged from 33 to 77 years-old (median age 67 and mean age 63). There were 35 WHO grade 1, 23 grade 2, and five grade 3 tumors. The histopathological characteristics of the tumors are shown in Figure 3A and the Appendix A. Genomic characterization was performed in all tumors using a gene panel including the 13 most frequently mutated genes in such tumors (see Material and Methods). Only “probably pathogenic” and “pathogenic” variants were considered. Overall, 29 tumors (46%) presented a *NF2* alteration (deletion, mutation or both; Figure 3B,C). The other genomic abnormalities were in the *TRAF7* (for 17.5%, 8/63), *AKT1*, *SMO*, *KLF4* or *CDKN2B* genes (Appendix A). No genomic abnormalities were found in 28.5% (12/63) of the tumors (Figure 3B). As expected, *NF2* alteration was more often present in WHO grade 2 and 3 tumors than in grade 1 (respectively 18/27 vs. 11/36, *p* = 0.01, Figure 3C) and in the skull convexity meningioma in comparison to the skull base ones (18/30 vs. 4/21 *p* = 0.0037). No canonical *BRAF*, *KRAS,* or *NRAS* mutation was observed in all tumors.

### 3.4. Drug Effects on Signaling Pathways on Meningioma in Primary Culture

The impact of known drugs was checked on protein targets of the Pi3K-AKT-mTOR and MAP kinase pathways in 4 meningiomas in primary culture by Western blotting (Appendix A, and Figure 4). Alpelisib inhibited the phosphorylation levels of AKT and of S6 (*p* < 0.05) and had no impact on the ERK pathway in the 4 tumors. Everolimus inhibited S6 phosphorylation (*p* < 0.05). As expected, trametinib fully inhibited the phosphorylation of ERK (*p* < 0.05) but, surprisingly, also increased the level of AKT phosphorylation (*p* < 0.05, Figure 4) as observed in the meningioma cell lines (Figure 1).

### 3.5. Drug Effects on Cell Viability on Meningioma in Primary Culture

The impact of alpelisib and everolimus was assessed on cell viability in vitro on 32 tumors randomly selected and including 20 grade 1, 9 grade 2, 3 grade 3 (Appendix A). Alpelisib and everolimus decreased cell viability of all meningiomas in a dose-dependent manner. No matter the grade (Figure 5A,B), the Ki67 index, the histopathological subtype or the mutation status, the same dose-dependent effect was observed. As described for other tumoral cells [22,23] a micromolar concentration of alpelisib was required in vitro to induce a decrease in cell viability of primary meningioma cells (mean IC_50_ 8.8 × 10^−6^ M) while a nanomolar concentration was sufficient for everolimus (mean IC_50_ 1.4 × 10^−10^ M). The mean maximal inhibition of cell viability was 94 ± 3% at 10^−4^ M for alpelisib and at 42 ± 2% at 10^−8^ M for everolimus.

The trametinib effect was assessed on cell viability on 42 randomly selected tumors and including 25 WHO grade 1, 15 grade 2 and 2 grade 3 (Appendix A). A dose response was observed in all tumors. The trametinib response level was weakly but significantly related to tumor aggressiveness, since the maximal inhibitory effect was lower in the 17 WHO grade 2 and 3 (35.8 ± 2.4%) than in the 25 WHO grade 1 (45.8 ± 2.5%, *p* < 0.01, Figure 5C). The IC_50_ was calculated as 2.2 × 10^−8^ M in WHO 1 tumors vs. 6.9 × 10^−8^ M in WHO 2 and 3 tumors. Moreover, the tumors with a Ki67 proliferative index >5% had a slight lower maximal inhibitory effect than tumors with a Ki67 ≤ 5% (mean inhibitory effect: Ki67 > 5% n = 24, 35 ± 4% vs. Ki67 ≤ 5% n = 18 49 ± 4%, *p* < 0.03, respectively; Figure 5D). No significant correlation was observed to the genomic status, particularly to *NF2* alterations, or to ERK phosphorylation levels as assayed by Western blot (Appendix A) on tumor fragments.

### 3.6. Mechanism of Cell Viability Inhibition on Meningioma in Primary Culture

In order to examine the mechanism of cell viability inhibition, we assessed, cell proliferation through BrdU incorporation and apoptosis through PARP cleavage as assessed by Western-blot or an increase in caspase 3/7 activities.

Seven tumors randomly selected were available for BrdU experiments (Appendix A). For all tested tumors, we found a clear dose-response effect under each drug (Figure 6A). The maximal inhibitory effect was −94.8 ± 10% under alpelisib and −50 ± 7.8% under everolimus. The IC_50_ were in the same range as those observed in the cell viability experiments for both of these drugs (4.39 × 10^−6^ M and 3.7 × 10^−10^ M, respectively). However, trametinib induced a stronger inhibitory effect on BrdU incorporation (maximal effect at −86 ± 6.6 %) than observed in the cell viability assay. Moreover, the IC_50_ (17.3 × 10^−9^ M) was lower in these experiments than in the former ones.

Apoptosis was assayed in 15 tumor primary cell cultures (Appendix A). Apoptosis was observed only under alpelisib treatment and in only 8 of 15 tumors (56%), through an increase of caspase 3/7 activities (mean 1.7-fold increase, *p* = 0.007 Figure 6D) and confirmed by the PARP cleavage for 4 tumors (*p* = 0.0286, Figure 6B,C). Alpelisib-induced apoptosis did not correlate with histopathological features, WHO grades, or assessed genomic abnormalities.

### 3.7. Effect of Combined Treatments Targeting Pi3K-Akt-mTOR and MAP Kinase Pathways on Meningioma in Primary Culture

We checked the impact of combined treatments, targeting both the Pi3K-AKT-mTOR and MAP kinase pathways on signaling pathway activity (Figure 4). AKT phosphorylation induced by trametinib or everolimus was reversed by alpelisib in the four human tested tumors (*p* < 0.05) whereas, and in contrast, was persistent with the combination everolimus and trametinib. Trametinib enhanced the inhibitory effect on S6 phosphorylation induced by alpelisib or everolimus (*p* < 0.05). Overall, the suppression by alpelisib of trametinib induced-AKT-phosphorylation in one side and the enhancement by trametinib of the alpelisib inhibition on S6 phosphorylation in the other side supported the interest of the combined treatment: alpelisib + trametinib (Figure 4).

For that, combined treatment by alpelisib (10^−6^ M) plus trametinib (10^−7^ M) was tested on the 3 cell lines and on 21 randomly selected meningiomas (14 WHO grade 1, 6 grade 2, and 1 grade 3, Appendix A). Inhibition of cell viability was significantly stronger in response to combined treatment than to each drug alone (mean percentage of inhibition 45 ± 3% under combined treatment vs. 15 ± 2% under alpelisib [*p* < 0.0001] or 31 ± 4% under trametinib [*p* < 0.0001], Figure 7A). This effect under combined treatment was additive between both drugs since the observed values were not statistically different from the calculated sums of effects (*p* = 0.16). The additive effect was also observed in WHO grade 2 and 3 tumors (Figure 7B).

No additive effect was observed between everolimus (10^−9^ M) and alpelisib ([10^−6^ M] (Appendix A). In contrast a significant stronger effect was observed under combined treatment everolimus and trametinib in comparison to each drug alone: mean percentage of cell viability inhibition: everolimus (10^−9^ M) 30 ± 6%, trametinib (10^−7^ M) 30 ± 8%, vs. combined treatment 52 ± 5% (*p* < 0.015 Appendix A).

## 4. Discussion

New targeted drug strategies in aggressive or recurrent meningiomas require additional understanding of intracellular signaling pathways.

Currently, the most common strategies used to treat aggressive, multi-recurrent meningiomas are the anti-VEGF, bevacizumab, and the combination of the SSTR2A inhibitor octreotide with everolimus [24]. Nevertheless, objective decreases in tumor mass are rare, a decrease in tumor growth rate or tumor stabilization are more frequent. Unfortunately, even these effects are inconstant, unpredictable, and mostly transient, over 1–3 years. Multiple chemotherapies and targeted therapies have been tested with disappointing results.

### 4.1. Pi3K-AKT-mTOR Pathway Targeting

The relevance of targeting the Pi3K-AKT-mTOR pathway in meningioma has by now been well demonstrated, particularly in the most aggressive meningiomas [7,8,9]. The link between activation of the Pi3K-AKT-mTOR pathway and *NF2* loss has been previously highlighted [6,9]. Our group has demonstrated that the mTOR inhibitor everolimus is efficient both in vitro and clinically to control cell proliferation and tumor growth of meningiomas [11,13,14]. In contrast, inhibitors targeting the Pi3K-AKT-mTOR pathway upstream of mTOR remain poorly studied in meningioma. This may be in part because the *AKT1* and *PiK3CA* mutations are rare, found only in 9% and 7% of WHO grade 1 meningiomas without *NF2* abnormalities [5,25]. In light of the multiple ways by which Pi3K-AKT-mTOR pathway activation impacts meningioma biology, we analyzed here for the first time the effect of the Pi3K inhibitor alpelisib and compared its effect with that one of the mTOR inhibitor, everolimus, commonly used today to treat aggressive meningioma. Alpelisib, also known as BYL719, is a selective inhibitor of the constitutively activated PI3Kα product of *PIK3CA* in response to either of two hot spot mutations: p.His1047Arg or p.Glu545Lys, both of which present in few meningiomas. Alpelisib also inhibits wild-type activity by inducing the degradation of PI3Kα [26]. This drug is considered to be one of the better clinically tolerated PI3K inhibitors and was the first orally available PI3K inhibitor approved by the U.S. Food and Drug Administration [26]. Several clinical trials are ongoing in multiple types of *PIK3CA*-mutated tumors, but also in *PIK3CA*-mutated vs. non-mutated breast cancers [15].

We have previously shown that everolimus induces AKT activation in meningioma cells, which may decrease its anti-tumoral activity [11]. Here, we demonstrate an inhibition of AKT phosphorylation in all the tested tumors. Moreover, an apoptotic effect was observed on approximately half of primary tumors (never observed with everolimus), and finally, a stronger inhibitory effect on cell viability as compared to everolimus in all cell lines and tumors. These effects were present no matter the WHO grades, histological subtypes or genomic status of the meningiomas tested.

Our results show that the Pi3K-AKT-mTOR pathway is homogeneously overactivated in meningioma cell lines and human meningioma and thus that the Pi3K-AKT-mTOR pathway is a crucial therapeutic target in such tumors [8,17]. Everolimus treatment is not sufficient to decrease tumor volume or to control tumor growth over the long term [13,14]. Thanks to efficient AKT inhibitory and additional apoptogenic effects, alpelisib is likely to exert stronger antitumoral activity in comparison to everolimus in the clinical setting. However, the IC_50_ values under alpelisib were higher than that of everolimus but were in the same range to what observed for other cell lines or primary tumor cells [22,23]. The issue is that the alpelisib IC_50_ is higher than the maximal achievable concentration in patient [23], predicting potential toxicity in patients receiving alpelisib in monotherapy at the concentration required for its antitumor activity [23].

### 4.2. MAP Kinase/ERK Pathway Targeting

The *BRAF*^V600E^ mutation is exceptional in meningioma [27], consequently only a few preclinical studies have explored the effects of MAP kinase pathway targeting in meningioma [8,16,17]. However, MAP kinase targeting could be of interest since the MAP kinase pathway is strongly activated particularly in low WHO grade meningiomas [8]. Trametinib is a reversible allosteric inhibitor of MEK1 and MEK2 activation. An inhibitory effect was observed with trametinib, slightly but significantly stronger in WHO grade 1 than in WHO grade 2 and 3 and, in agreement, in tumors with Ki67 < 5% than in those with Ki67 > 5%. No correlation was observed with mutational status or ERK phosphorylation levels. When targeting MAP kinase signaling with trametinib, we found no apoptosis, in contrast to a 2005 study by Mawrin et al. [8]. However, here they had analyzed only one malignant meningioma and the MAP kinase inhibitor used in this study was PD98059, a less selective MEK inhibitor. Our results demonstrated a strong antiproliferative with a IC_50_ using brdU incorporation higher than one obtained by cell viability assay. Our data suggest that trametinib could be an interesting therapeutic option particularly in slow or intermediary growing WHO grade 1 meningothelial meningiomas despite radiation therapy and non-accessible to surgery as seen in some skull base ones.

### 4.3. Co-Targeting the Pi3K-AKT-mTOR and MAPkinase Pathways

The Pi3K-AKT-mTOR and MAP kinase signaling pathways are activated in a wide range of tumors, including meningiomas. We have observed that MEK inhibition leads to PI3K/AKT activation in meningioma cells, as previously shown in breast cancer cells [28]. Moreover, co-targeting Pi3K-AKT-mTOR and MAP kinase pathways with a combination of alpelisib and trametinib provided an additive effect on the inhibition of cell viability in Ben-Men-1, IOMM-Lee and all human fresh meningiomas tested. Moreover, alpelisib reversed the AKT phosphorylation induced by trametinib and trametinib enhanced the S6 phophosphorylation induced by alpelisib.

A strong inhibitory effect was also observed with the combined everolimus plus trametinib treatment as compared to each drug alone. However, this combined treatment also induces an increase in trametinib-induced AKT phosphorylation that could be a negative point in this therapeutic strategy. Redundant inhibition of the Pi3K-AKT-mTOR pathway using both alpelisib and everolimus is less interesting. Overall, our data support the clinical interest of combining both alpelisib and trametinib molecules in treating meningiomas.

### 4.4. Relevance and Limits of this In Vitro Study

In a first step, the drugs were assessed on 3 human meningioma cell lines, 2 with *NF2* deficiency (Ben-Men-1, CH-157MN) and one without (IOMM-Lee). Among the 3 cell lines, the behavior of Ben-Men1- is closest to those of human fresh tumors, probably due to the accumulation of oncogenic events as *BRAF* or *NRAS* mutation in the 2 others CH-157MN and IOMM-Lee. We have previously demonstrated that meningioma primary cell cultures are a highly relevant model to study targeted therapies, decipher molecular mechanisms and establish proof-of-concept to set up clinical trials [11,12,13,14]. We have validated this preclinical meningioma model in our study of the combined treatment octreotide and everolimus [11,12] that later led to the CEVOREM clinical trial [13]. The anti-proliferative effect observed in vitro indeed led to a tumoral growth rate decrease in vivo in patients [11,13,14]. To that purpose, we worked with freshly operated meningiomas to preserve membrane receptor expression in addition to intracellular signaling pathway activation and never with frozen cells. A wide range of meningiomas was included to check several hypothetical correlations with clinical data. However, this in vitro model does not take in consideration the impact of microenvironment on cell responses. Moreover, the number of experimentations performed on each tumor was time-limited (2–3 weeks after surgical removal) and was also limited by the cell tumoral quantity.

## 5. Conclusions

We have demonstrated that targeting the Pi3K-AKT-mTOR pathway is relevant in all meningiomas; regardless of any mutations of *Pi3K* or *AKT* components but also *NF2* status. Moreover, alpelisib induced a stronger inhibitory effect in comparison to everolimus with a proapoptotic effect in 50% of tumors. We demonstrated that MEK inhibition decreases cell proliferation, slightly stronger in WHO grade 1 than WHO 2 and 3. Other pathways, such as the Hippo signaling cascade, may be involved in such variable responses [29]. Alpelisib treatment should be probably only considered in a combined therapies [23]. In that way, the combination of alpelisib and trametinib showing an additive inhibitory effect could present a new therapeutic strategy for recurrent or high-grade meningiomas. However, the clinical tolerance of such a combination should be determined.

## Figures and Tables

**Figure 1 cancers-14-04448-f001:**
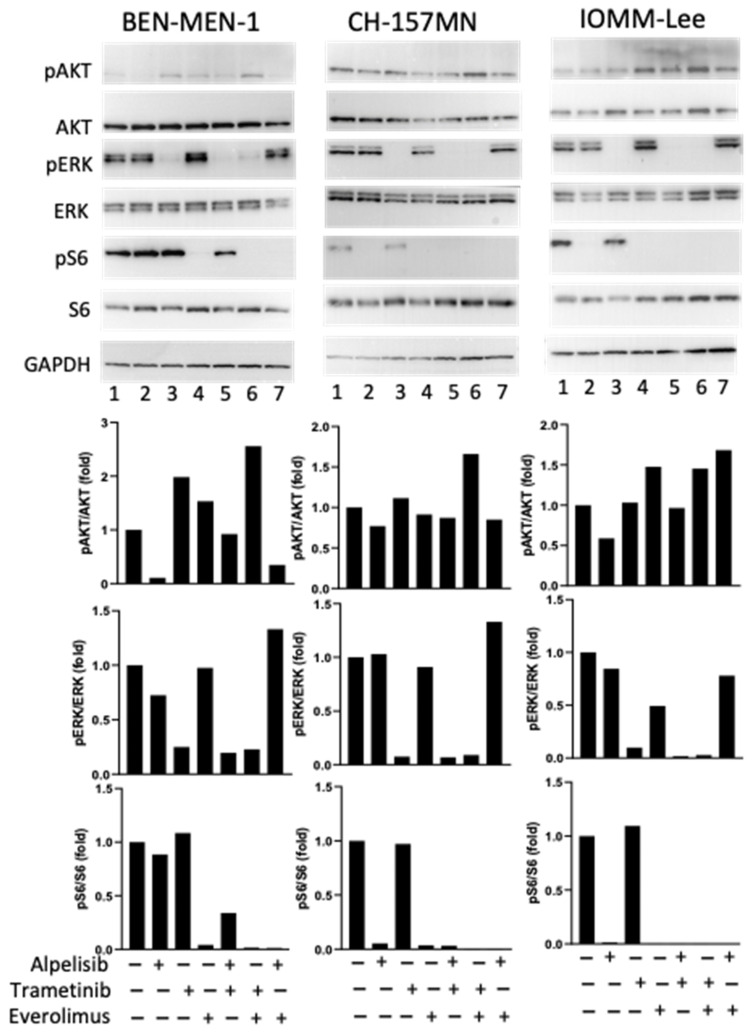
Alpelisib, trametinib and everolimus on the MAP kinase and Pi3Kinase of 3 meningioma cell lines: Ben-Men1, CH-157MN and IOMM-Lee. Representative Western blot showing phospho-AKT on Ser473 (pAKT), total AKT, phospho-ERK1/2 on Thr202/Tyr204 (pERK) and total ERK1/2, phospho-S6 on Ser235/236 (pS6) and S6 and GAPDH after 1.5 hours’ incubation of meningioma cells with alpelisib (10^−6^ M), trametinib (10^−7^ M) or everolimus (10^−9^ M) alone or in combined treatment in comparison to untreated cells. “+”: with the drug; “−”: without the drug. The original blots are shown in Appendix A.

**Figure 2 cancers-14-04448-f002:**
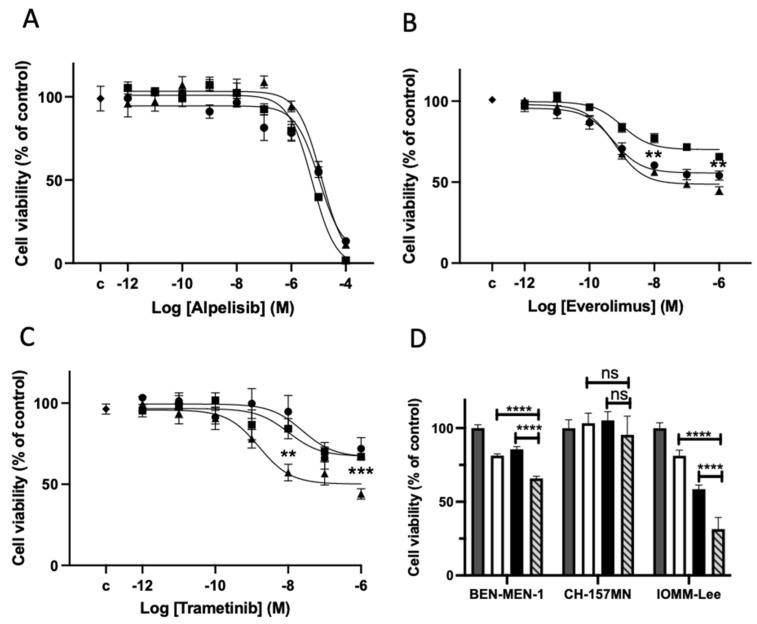
Alpelisib, trametinib and everolimus alone (**A**–**C**) or in combined treatment (**D**) on the cell viability of the 3 meningioma cell lines: Ben-Men1 (black round), CH-157MN (black square) and IOMM-Lee (black triangle). (**A**–**C**): Dose-effect curve under treatment for 3 days. Each experiment was done 3 times in triplicate. Results are expressed as mean ± SEM percentage of cell viability in comparison to DMSO (dimethylsulfoxide) treated control cells (“c”). (**B**): ** *p* < 0.01 between the everolimus inhibitory effect on CH-157MN vs. that on Ben-Men1 or on IOMM-Lee at 10^−8^ M or 10^−6^ M. (**C**): *** *p* < 0.001 between the trametinib inhibitory effect on IOMM-Lee vs. that on Ben-Men1 or on CH-157MN at 10^−8^ M. ** *p* < 0.01: the same at 10^−6^ M; (**D**): Combined treatment of alpelisib 10^−6^ M with trametinib 10^−8^ M (hatched bar) in comparison with alpelisib 10^−6^ M alone (white bar) and with trametinib 10^−8^ M alone (black bar) at 3 days. Results are expressed as mean ± SEM percentage of 3 experiments in triplicate, of cell viability relative to DMSO-treated control cells (grey bar). ns: non-significant. **** *p* < 0.0001.

**Figure 3 cancers-14-04448-f003:**
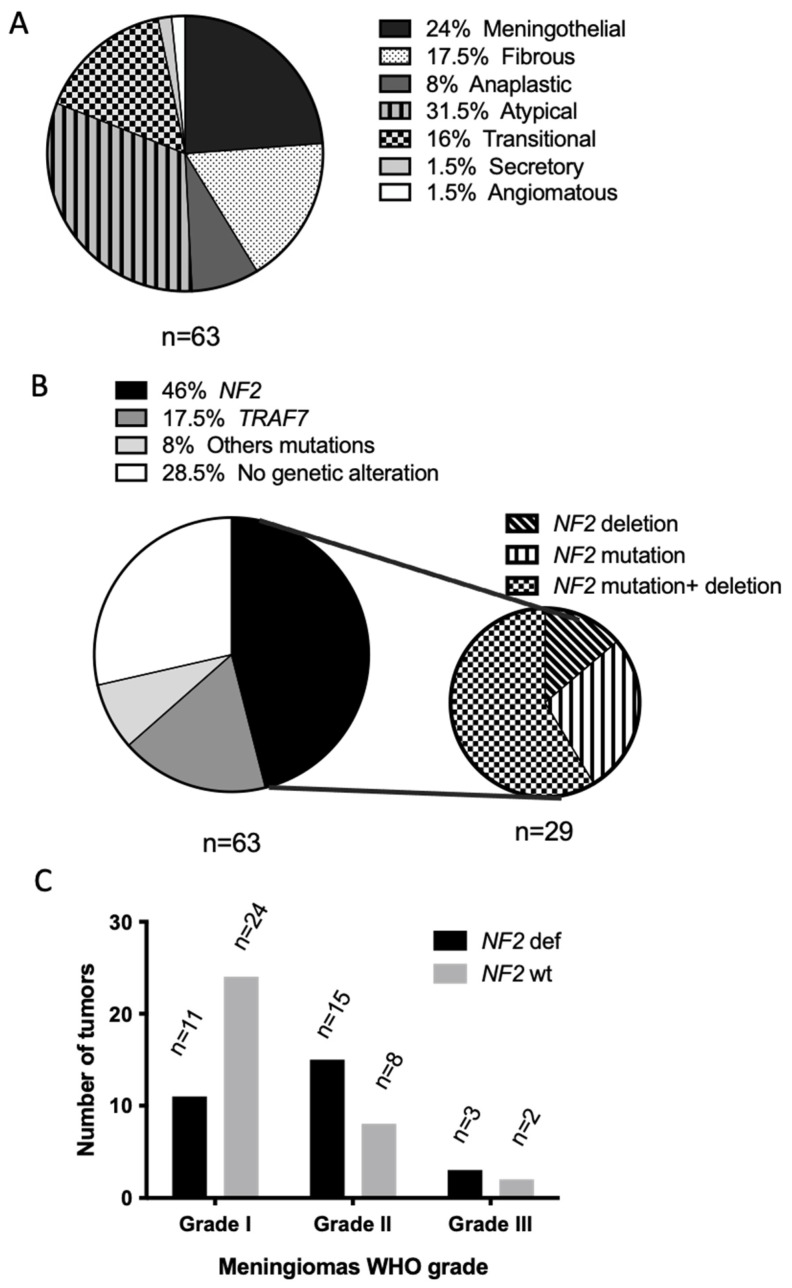
Histopathological (**A**), genomic (**B**) and 2021 WHO grade (**C**) characteristics of the 63 meningiomas. *NF2* wt: *NF2* wild-type; *NF2* def: *NF2* deficiency meaning tumors with *NF2* alterations, mutation, deletion or both.

**Figure 4 cancers-14-04448-f004:**
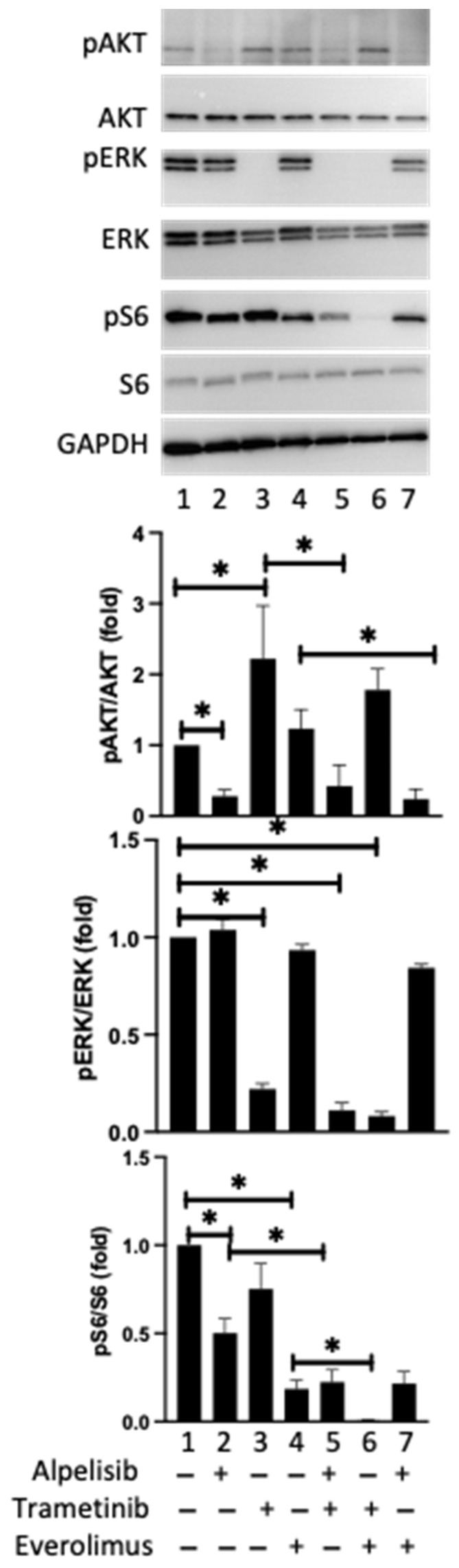
A representative Western blot in one meningioma (M55) showing phospho-AKT on Ser473 (pAKT), total AKT, phospho-ERK1/2 on Thr202/Tyr204 (pERK) and total ERK1/2, phospho-S6 on Ser235/236 (pS6) and S6 and GAPDH after 3 h’ incubation of meningioma cells with alpelisib (10^−6^ M), trametinib (10^−7^ M) or everolimus (10^−9^ M) alone or in combined treatment in comparison to untreated cells from one meningioma. Quantification of pAKT/AKT, pERK/ERK and pS6/S6 ratio in 4 randomly selected meningiomas (M55, M60, M61, M63). The results are represented as mean fold change ±SEM relative to untreated cells. “+”: with the drug; “−”: without the drug. *: *p* < 0.05. The original blots are shown in Appendix A.

**Figure 5 cancers-14-04448-f005:**
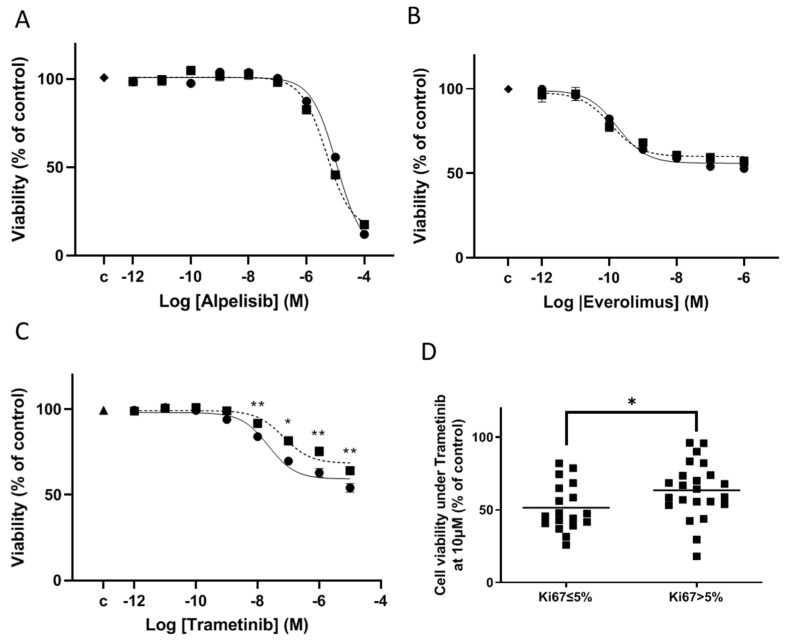
Dose-effect curve on cell viability under treatment with alpelisib (**A**), everolimus (**B**) or trametinib (**C**,**D**) for 3 days, quantified using a cell viability assay. Results are expressed as mean ± SEM percentage of cell viability in comparison to DMSO (dimethylsulfoxide)-treated control cells (“c”). Overall, 32 tumors, randomly selected were tested after administration of alpelisib and everolimus (mean inhibitory effect on 20 grade 1 tumors [continuous line] vs. mean inhibitory effect on 12 grade 2 or 3 tumors [dotted line]), and in (**C**) 42 after trametinib (mean inhibitory effect on 25 grade 1 [continuous line] vs. mean inhibitory effect on 17 grade 2 or 3 [dotted line], ** *p* < 0.01: comparison of trametinib inhibitory effect in the 25 grade 1 tumors vs. in the 17 grade 2 and 3 tumors at each concentration). In (**D**), the effect on cell viability of 10^−6^ M trametinib in the 42 tumors classified according to their Ki67 index (less than or equal to 5% and greater than 5%). Black line: mean Ki67 index. * *p* < 0.05.

**Figure 6 cancers-14-04448-f006:**
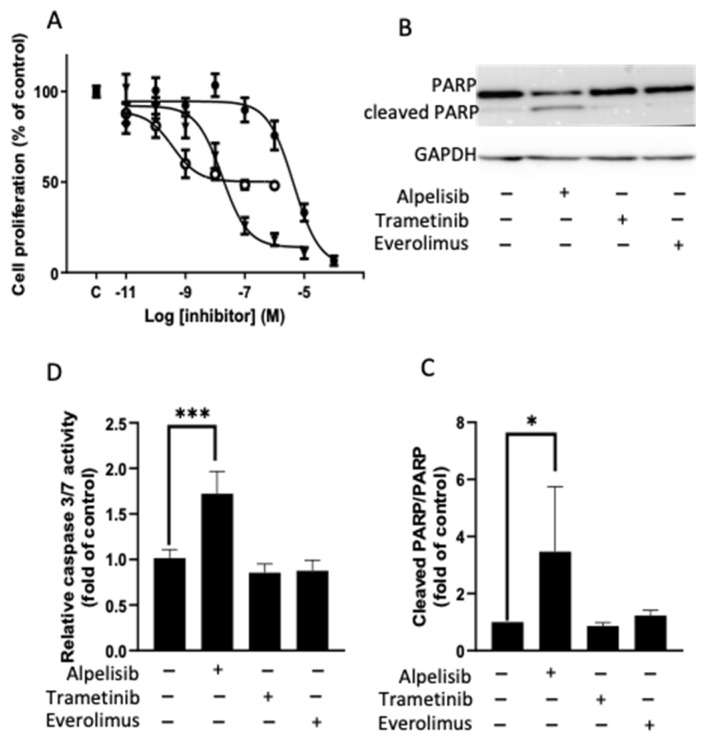
Mechanism of cell viability inhibition. (**A**): cell proliferation measured by BrdU incorporation of 7 randomly selected meningiomas (see Appendix A) after 2 days’ of incubation period with an increasing dose of alpelisib (black points), trametinib (black triangles) and everolimus (white circles). Results are expressed as mean ± SEM percentage of BrdU incorporation in comparison to the control condition (“c” black square). PARP cleavage (**B**,**C**) and caspase 3/7 activities (**D**) in tumors in which alpelisib induced apoptosis. (**B**): a representative western-blotting of PARP cleavage on one meningioma (M8) after 2 days’ incubation period with alpelisib 10^−6^ M, trametinib 10^−7^ M, or everolimus 10^−9^ M. (**C**): the mean quantification of PARP cleavage on 4 meningiomas (M8, M11, M18, M30). Results are expressed as mean fold change ± SEM relative to control conditions * *p* < 0.05. (**D**): caspase 3/7 activity, after 2 days’ incubation period with alpelisib 10^−6^ M, trametinib 10^−7^ M or everolimus 10^−9^ M on M18, M24, M32, M33. *** *p* < 0.001 Results are expressed as mean fold change ± SEM relative to control cells. “+”: with the drug; “−”: without the drug. The original blots are shown in Appendix A.

**Figure 7 cancers-14-04448-f007:**
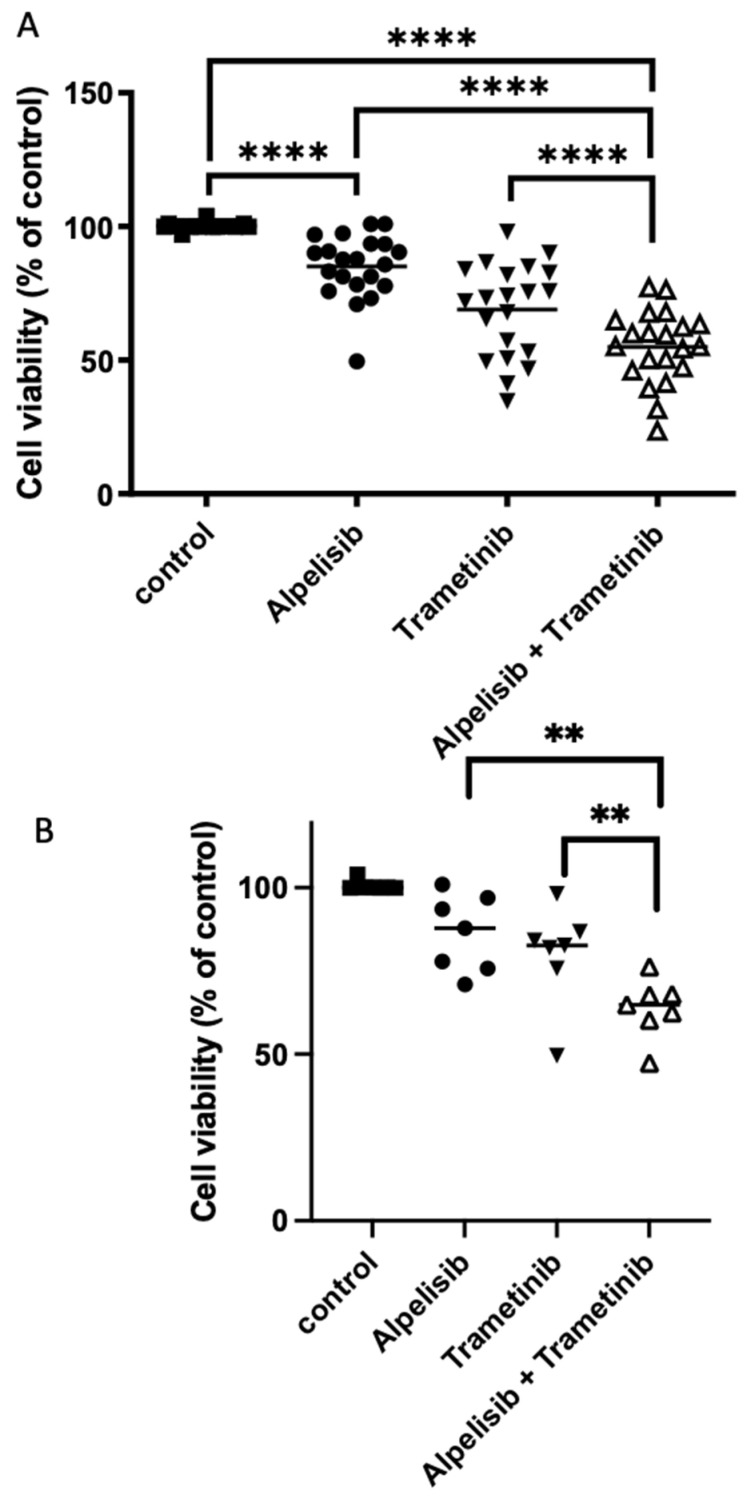
Combined treatment of alpelisib and trametinib in meningioma. (**A**): Inhibitory effect on cell viability of 10^−6^ M alpelisib, 10^−7^ M trametinib or combined treatment of both drugs on 21 meningiomas, including 14 WHO grade 1, 6 grade 2 and 1 grade 3 primary tumor cells. For each tumor, results are expressed as the percentage of cell viability relative to DMSO (dimethylsulfoxide)-treated control cells (“control”). (**B**): the same on 7 WHO grade 2 and 3 tumors. Black line: mean reduction of cell viability for the tumors. **** *p* < 0.001; ** *p* < 0.01.

## Data Availability

The data can be shared up on request.

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
