# Peer review of "Co-Targeting MAP Kinase and Pi3K-Akt-mTOR Pathways in Meningioma: Preclinical Study of Alpelisib and Trametinib"

_cancers, 2022, doi:10.3390/cancers14184448_

Round 1

Reviewer 1 Report

Dear authors

This research is nicely done with comparison of cell lines of different genetic mutation and clinical tumor cells. You found that MAPK and PI3K-AKT-mTOR pathways are critical in meningioma cell proliferation from the cell line and clinical tumor data (according to the NF2 deletion). You also chose the good selective pathway inhibitors for the research and found interesting facts that the Trametinib and Alpelisib could be the best combo overall as they do not accumulate negative effects. 

1. The Language has to be improved. There are some grammar and spelling errors too. Punctuations and citation brackets are missing in some sentences. Please carefully check and correct them.

2. section 3.1 please make a figure if the real data is not capable to show.

3. Figure 4 needs to do quantification (statistical significance) analysis.

4. figure 5D the labels of X-axis should be ki<5% and ki>5%. 

5. It is always nice to combine targeting therapy drugs if the pathways are critical, but it is also important to investigate the toxicity of the combination over normal cell. It would be very informatic if normal cell viability is checked with the combination treatment.

6. According to the figure Why don't you investigate the PARP cleavage over the combination treatment? In addition, if you are looking into apoptosis, you could also investigate the cell cycle arrest in flow cytometry.

Overall this is a novel research with clear experimental setup and rational choose of combination treatments based on experiment results. 

Reviewer 2 Report

Comments to Author

Mondielli et al, examined Alpelisib (P13K inhibitor) and Trametinib (MEK inhibitor) in vitro in various meningioma cell lines to study the cell viability as well as to study the PI3K- AKT-mTOR and MAP kinase pathways, which are known to be overactivated in major meningioma cases.

Overall work is significant since they studied the most common and rare to treat type of cancer, meningiomas. There is need to always develop new therapies and their work shows promising new therapeutics that can be used in combination with everolimus (known mTOR inhibitor). The below points need to be addressed.

1.      There are numerous formatting issues including punctuations, spaces between words (so many), representation of concentrations (check superscripts and subscripts- Eg: 10-6M , CO2 etc.), numerous grammatical errors that will make the manuscript clear and easy to understand. Check through out the paper starting from Abstract till the conclusion page. Look below for some of them.

2.      In methods,

a) Line 109-110: Why 39% DMEM was used? Any specific reason or that is the manufacturers available concentration. If so, mention the company from which the media was obtained.

b) Lines 110-111: Check space and subscripts for CO2.

c) Line 148- space between NF2  and biallelic

d) Line 155- Correct the superscript 2 x 104 cells.

e) Line 178- Is it 5 x 105 cells?

f) Line 185- 10,000xg (check punctuation)

3.      Results and Figures:

a)      Lines 223-224, show the data in the supplementary

b)      Figure 1: It is good to have the lanes numbered so it is easier to compare. Also, add the quantification data for easier comparison. Since GAPDH looks darker in some lanes compared to others indicating, it looks like there is no equal protein concentration in lysates.

c)      Lines 242: This looks like a very high concentration of drug? How did you choose it?  or is it formatting issues and is in micromolar or millimolar range? Correct the numeric.  Is it 9.6 micromolar?? Check them all throughout.

d)      Figure 2: The graphs need to be much clear. Except the Alpelisib (Fig 2A), the others don't seem to be completing the dose response curve indicating, the dose used is not enough to show IC50 values. How did you get the values of IC50? Line 267- cannot be concluded since the graphs are not clearly representing for other drugs.

e)      Line 289: Format spacing between “pathways in 4”

f)       Figure 4:  Same thing as in previous western blot. Inconsistent GAPDH bands. The results cannot be trustworthy if the control is bad. Show statistical significance in the graphs if any, between different treatment options.

g)      Lines 309-313: Please check these values. the graph doesn't seem to representing accurately because of incomplete curves. Indicate how these are calculated.

h)      Lines 331-333 : Please include it in supplementary figures.

i)       Line 345-346: Any reason behind this lower IC50 values? Also check the superscript in concentration (10-9)

j)       Figure 6: Western blot image in 6B needs to be quantified.

4.      Discussion: Formatting issues again.

a)      Line 427-  what context is it?

b)      Line 447- Please clarify the continuity of the statement

Round 2

Reviewer 2 Report

The authors have answered and corrected all my comments.